# Ursolic Acid and Its Derivatives as Bioactive Agents

**DOI:** 10.3390/molecules24152751

**Published:** 2019-07-29

**Authors:** Sithenkosi Mlala, Adebola Omowunmi Oyedeji, Mavuto Gondwe, Opeoluwa Oyehan Oyedeji

**Affiliations:** 1Department of Chemistry, Faculty of Science and Agriculture, University of Fort Hare, Private Bag X1314, Alice 5700, South Africa; 2Department of Chemical and Physical Sciences, Faculty of Natural Sciences, Walter Sisulu University, Private Bag X1, Mthatha 5117, South Africa; 3Department of Human Biology, Faculty of Health Sciences, Walter Sisulu University, Private Bag X1, Mthatha 5117, South Africa

**Keywords:** non-communicable diseases, pentacyclic triterpenoids, ursolic acid, derivatives, sources, biological studies, clinical trials.

## Abstract

Non-communicable diseases (NCDs) such as cancer, diabetes, and chronic respiratory and cardiovascular diseases continue to be threatening and deadly to human kind. Resistance to and side effects of known drugs for treatment further increase the threat, while at the same time leaving scientists to search for alternative sources from nature, especially from plants. Pentacyclic triterpenoids (PT) from medicinal plants have been identified as one class of secondary metabolites that could play a critical role in the treatment and management of several NCDs. One of such PT is ursolic acid (UA, 3 β-hydroxy-urs-12-en-28-oic acid), which possesses important biological effects, including anti-inflammatory, anticancer, antidiabetic, antioxidant and antibacterial effects, but its bioavailability and solubility limits its clinical application. *Mimusops caffra*, *Ilex paraguarieni,* and *Glechoma hederacea,* have been reported as major sources of UA. The chemistry of UA has been studied extensively based on the literature, with modifications mostly having been made at positions C-3 (hydroxyl), C12-C13 (double bonds) and C-28 (carboxylic acid), leading to several UA derivatives (esters, amides, oxadiazole quinolone, etc.) with enhanced potency, bioavailability and water solubility. This article comprehensively reviews the information that has become available over the last decade with respect to the sources, chemistry, biological potency and clinical trials of UA and its derivatives as potential therapeutic agents, with a focus on addressing NCDs.

## 1. Introduction

Non-communicable diseases (NCDs), including cancers, cardiovascular diseases, pulmonary illnesses, and diabetes, account for nearly 71 percent of all fatalities globally, as reported by the World Health Organization. The rapid increase of NCDs has been influenced by a series of flexible behavioural threat variables, including unhealthy eating habits, physical inactivity, cigarette smoke exposure and damaging use of alcoholic drinks, as well as natural air pollutants, metabolic (hypertension, obesity, hyperglycaemia, hypercholesterolemia) and occupational (gases, carcinogens, fumes, particulates) risks [1].

On the other hand, natural products have been recognized in therapeutic drugs for many years as an essential source of active substances. Therapeutic agent complexity depends on synthetically prepared chemical combinations. Therapeutic drugs are regarded to be natural, synthetic or semi-synthetic, depending on the source from which they have been derived. Natural therapeutic agents are made from naturally occurring compounds that contain active ingredients extracted from sources such as plants [2]. One group of compounds from plants is called pentacyclic triterpenoids (PT). These are defined as a group of 30 carbon structural compounds with interesting biological activities and great diversity, with more than 100 chemical skeletons of carbon. PT, which have been studied for over two decades, with a perceptible increase in their pharmacological significance, are described to have strong health benefits, including hepatoprotective, wound healing, antibacterial, anti-inflammatory, antitumoral, and antiviral properties, together with their low toxicity [3]. PT are used in treatment of several cancers without damaging normal cells through toxicity [4]. These PT have also revealed therapeutic and chemical preventive characteristics in skin, lung, liver and lung disease [5]. The consumption of vegetables and fruit by animals is thought to have a reduced incidence of cancer and several illnesses. PT have been separated and elucidated from various plants [6]. Oleanolic acid (OA), ursolic acid (UA) (**1**), betulinic acid (BA), and lanosterol are examples of PT [7,8]. Triterpene groups, along with their classified chemical compounds and their characteristics, are shown in Table 1. For interesting chemical structures of pentacyclic triterpenoids such as UA, their reactivity and interactions with a disease are essential for searching through effective drug sites. Many triterpenes can be used as bioactive compounds immediately or modified to enhance their potency and selectivity [9]. Therefore, this review provides an overview of the structural development of UA and its derivatives, along with their advantages as anti-cancer, anti-inflammatory, antibacterial, anti-diabetic and neuroprotective therapeutic agents, as well as their herbicidal activities.

## 2. Chemistry of UA

Ursolic acid (3 β-hydroxy-urs-12-en-28-oic acid) is a pentacyclic triterpenoid composed of a C-30 chemical structure built from isoprenoid units with A, B, C, D and E rings (Figure 1) [51]. UA may occur as an aglycone or free acid of saponins [52,53]. UA has a low solubility in water, but high solubility in alcoholic sodium hydroxide (NaOH) and glacial acetic acid [54,55,56,57]. The UA chemical formula is C_30_H_48_O_3_, with a melting point of 283–285 °C. The properties of the UA structure indicate that it has a molecular weight of 456.7g/mol [20,52,53,58]. UA is mainly biosynthesized by folding and cycling squalene from a dammarenyl cation, which makes the third UA ring by expanding the ring and creating an added ring [58]. The biosynthesis of UA found in plant cells originates from the cyclical (3*S*)-oxidosqualene cycling. The predominant (3*S*)-oxidosqualene predecessor is converted into a cationic dammarenyl structure undergoing chain growth and other cyclic changes in order to form this third distinct ring, which is present in the α-amyrin skeleton, a nucleus in the UA [51].

It is well known that low aqueous solubility of a drug may seriously affect its medication effectiveness. It has also been documented that certain treatments have a low solubility and few side effects. Ursolic acid, however, has low water solubility, which limits bioavailability in the human body [59]. The limited solubility, poor bioavailability and rapid metabolism of UA have limited their additional clinical applications for therapeutic effect in numerous diseases. Hence, many researchers in recent years have reported various modifications in an attempt to improve UA pentacyclic triterpenoids’ potency, solubility and bioavailability in water [59,60]. Hence, this review focuses on some of the recent modifications to UA aiming at their use as bioactive agents with promising therapeutic effects of non-communicable diseases.

The chemical modifications to date have focused mainly on the hydroxyl group at position C-3, the unsaturated double bond at position C12-C13, and the carboxylic acid at position C-28 at rings A, C and E, respectively, as displayed in Figure 1 [61]. Mostly, structural modifications were made to above-mentioned positions in an effort to enhance the potency and bioavailability of UA and its derivatives, and study their structure-activity relationships and mechanisms [62]. For this reason, researchers started the search for new novel derivatives of through modifications of chemical groups in UA. Shao and colleagues (2011) in their UA (**1**) structure-activity relationship synthesized a total of twenty-three analogues through modification at position C-3 and C-28. 3-*O*-acetylursolic acid was obtained from acetylation of UA, which was later treated with bromo-diolefine to yield fatty esters as shown in Figure 2. They further synthesized UA amides and ester derivatives (Figure 3) [60]. 

Batra and Sastry (2013) reported isolation of UA (**1**) from *Ocimum sanctum*, which led to the preparation of three derivatives, namely, 3β-acetoxy-urs-12-en-28-oic acid (**25**), [*N*-(3β-acetoxy-urs-11-oxo-12-en-28-acyl) aniline] (**26**) and [Methyl *N*-(3β-butyryloxyl-urs-12-en-28-oyl)-2-amine acetate] (**27**) as shown in Figure 4 [63].

Nascimento and colleagues (2014) semi-synthesized two analogues of UA isolated from the Sambucus australis plant. Modifications to the UA structure were made at C-3 (hydroxyl group) in order to yield 3β-formyloxy-urs-12-en-28-oic acid (**19**) and 3β-acetoxy-urs-12-en-28-oic acid (**20**) [64].

In order to examine the effective locations, Meng and collaborators (2017) discovered a number of exciting ursolic UA derivatives. They developed and evaluated 11 derivatives (**30**–**41**) (Figure 5, Figure 6, Table 2). They found that the inclusion of an acetyl group at C-3 and an amino alkyl at C_28_ increased biological potential [65].

Sahni and others (2016) reported the isolation of UA from acetone extract of a *Eucalyptus hybrid*. In an attempt to increase the potency of UA, these authors synthesized a total of 6 compounds (esters and amide) at positions C-3 and C-28 (Figure 7). In the preparation of methyl (**42**), ethyl (**43**) and propyl esters (**44**), UA was treated with acetic anhydride, butyril chloride, and propyl chloride at position C-3 respectively in the presence of DMAP and THF catalysts. C-17 methyl ester (**45**) was developed by treating UA with methyl iodide and modifying the C_28_ carboxylic acid group. A C-17 propyl amine (**46**) variant was prepared by the oxalyl chloride treatment of UA in the presence of CH_2_Cl_2_ and then propyl amine [66].

Wu and colleagues (2017) produced several UA derivatives (**47**–**61**) in an effort to study the greater power and bioavailability of the UA by adding an acyl group at position C-3 to make the most of the structure-activity relationship. They further synthesized ester derivatives by a process of esterification with suitable acid chlorides in the presence of a DMAP catalyst. In addition, Claisen Schmidt condensation and Jones oxidation steps at positions C-2 and C-3, respectively, were used for bioactive UA derivatives (Figure 8) [62].

In a separate study, Wu and colleagues (2015) synthesized and elucidated several UA derivatives (**62–78**) as displayed in Figure 9. These researchers explored the structure-activity relationship of the synthesized bioactive analogues. They mainly focused their modifications on hydroxyl position C-3 and carboxylic group C-17 of the UA parent compound to produce a series of derivatives. In the first step, 3-*O*-acetate derivative was developed by the addition of acetic anhydride in anhydrous pyridine, which was later treated with oxalyl chloride to yield the 28-acyl-chloride intermediate. They further dissolved the intermediate in dichloromethane to give a series of amino compounds including aminobenzene, p-fluoroaniline, p-chloroaniline, p-bromobenzenamine, p-methoxylaniline o-fluoroaniline, o-chloroaniline and o-bromobenzenamine through condensation in the presence of triethylamine. After this, saponification analogues (**62**–**69**) were produced, which were later hydrolysed to give more derivatives (**70**–**77**). Lastly, the 3-oxo anlogue (**78**) was developed through oxidation with pyridinium chlorochromate. All the structure of these UA derivatives were elucidated by a series of spectroscopic techniques, such as nuclear magnetic resonance (NMR) including ^13^C-NMR and ^1^H-NMR with melting point, high resolution mass spectrometry, and electrospray ionization mass spectrometry (ESI-MS) [67].

Several UA-derived compounds have been developed with the aim of improving their potency and selectivity. Gu and others (2017) synthesized an interesting series of UA derivatives containing oxadiazole and quinoline chemical groups. One analogue is 3-oxo-ursolic acid (**78**) (78% yield), which was prepared by dissolving UA in acetone solvent and later oxidized using Jones reagent, focussing on position C-3. More derivatives (**79**–**82**) (62–68% yield) were synthesized by reacting 3-oxo-ursolic acid with corresponding o-aminobenzaldehyde under nitrogen (N_2_) molecule atmospheric conditions, as shown in Figure 10. Compound (**79**) was synthesized by reacting 3-oxo-ursolic acid with o-aminobenzaldehyde according the Friedlander reaction. Afterwards, 28-acylchloride derivatives were prepared by treating compound (**79**–**82**) with thionyl chloride. Furthermore, 28-acylchloride derivatives were treated with aryl hydrazine in the presence of trimethylamine (EtN_3_) to yield **83**–**94** (47–48% yield). Meanwhile, the dehydration condensation of **83**–**94** yielded oxadiazole derivatives (**95**–**106**) (56–74% yield), as shown in Figure 10. All of these compounds were purified through column chromatography and elucidated by different spectroscopic techniques including infrared (IR), ^13^C-NMR, ^1^H-NMR, ESI-MS and elementary analysis [68].

Many researchers, in search of more bioactive compounds, have reported different UA derivatives. Herewith are some of UA analogues with interesting mechanisms related to currently existing non-communicable diseases utilizing positions C-2, C-3, C-20 and C-28 (Figure 11) [69,70,71,72,73]. One UA derivative compound, **118** (2α,3β,7β,23-tetrahydroxyurs-12-ene-28-oic acid) (Figure 11), is a naturally occurring compound isolated from *Castanea crenata* Sieb. et Zucc. This compound was elucidated using several spectroscopic techniques, including IR, ^1^H-NMR, ^13^C-NMR, and HR-ESI MS [74].

## 3. Sources of UA and Its Biological Potency

Ursolic acid is a five-membered ring extensively distributed in food, medicinal herbs, fruit, vegetables, etc. [19,75,76,77,78,79,80,81,82]. UA triterpenoid is well documented to be available in fruits such as cranberries (*Vaccinium macrocarpon*) [53,83], blueberry (*Vaccinium* spp.) [77], basil (*Ocimum basilicum*) [78,84], olive (*Olea europaea*) [20], heather flower (*Calluna vulgaris*) [79], pear (*Pyrus pyrifolia*), Labrador tea (*Ledum groenlandicum* Retzius) [77], rosemary (*Rosmarinus officinalis*) [65] and apple peels, which possess important health benefits. UA is also distributed among higher plants, such as *Mimusops caffra, Ilex paraguarieni, Glechoma hederaceaes, Ichnocarpus frutescens*, and *Syzygium claviflorum* [19,77,80]. More plants sources, quantities of UA and its biological applications are hereby reported in Table 3.

UA has been investigated and is reported to possess many health benefits, including anti-apoptotic, anti-carcinogenic, anti-inflammatory, antioxidant, antirheumatic, antiviral, antitumoral, trypanocidal, etc. [61,62,78]. UA also has health benefits, and there are many reports of its anticancer activity in several cancers, including breast, skin, lung, prostate and pancreatic cancers [19,20,63]. Due to its low toxicity, anticancer activities, and commercial availability with various structural modifications, UA is regarded as a pillar through organic semi-synthesis, and this has attracted more researchers to studying and discovering various ursolic acid derivatives [61]. UA is popular because of its antiproliferative properties, inducement of cancer apoptosis, prevention of tumorigenesis, and/or blocking of the cell cycle in cancer cells. This was evident in the observation from one apoptosis mechanism which showed UA to have the ability to prevent Nuclear Factor κB (NF-κB) pathway by p65 phosphorylation suppression, resulting in mandatory decrease in various downstream oncogenes such as B-cell lymphoma-extra-large (Bcl-XL) and B-cell lymphoma-2 (Bcl2). Nonetheless, the antitumor potency of UA is poor because of its lower solubility, which decreases the drug absorption in the human system, leading to challenges in obtaining its full benefits. Therefore, it is necessary to construct its derivatives through semi-synthetic modifications in order to improve its antitumor activity. Mostly, the modifications of ursolic acid occur at sites C-3, C12-C13, and at position C-28 [59,63,64,72]. Jiménez-Arellanes and colleagues (2013) reported antimicrobial activity of pentacyclic triterpenoid compounds, namely, UA against *Mycobacterium tuberculosis* H37Rv [85]. Ursolic acid has been investigated in different stages of clinical trials for its therapeutic effects and selectivity against a diversity of diseases [86].

## 4. Biological Effects and Clinical Trials of UA and Some Derivatives

### 4.1. Anti-Inflammatory

Pathogenesis and homeostasis are part of inflammation. The inflammatory response is initiated after harm and/or microbial invasion to recover homeostatic tissue balance between composition and physiological function. Persistent inflammation may lead to damage of tissues, resulting in non-functioning organs [93]. Inflammation is a complicated occurrence linked to the development of different diseases, such as cardiovascular and neurodegenerative diseases, and cancer [55]. Rudolf Virchow proposed the link between inflammation and cancer as early as 1863. Currently, acute inflammation, with concomitant cytokine activity and enhanced output of reactive oxygen species, is identified as a cancer-promoting disease [3,16]. Zerin and co-authors (2016) examined the capacity of UA to decrease the production of TNF-α in RAW 267.4 and A549 cells infected with *Mycobacterium tuberculosis* and Con A-stimulated mouse splenocytes to detect anti-inflammatory activity. TNF-α is pivotal to inflammation and its impact on IL-1β and IL-6 decrease has been examined under comparable circumstances of therapy. These authors also studied UA activity with the aim of decreasing the levels of inflammatory intercessor, cyclooxygenase 2 and NO synthase found in stimulated cells. UA exhibited significant inhibitory effects on cytokine expression levels, immunomodulatory mediators, and release of NO. It is suggested that this compound can be used for tuberculosis and antibiotic therapy because of the UA anti-inflammatory potency in the mentioned cells [94]. Huang and others (2016) reported a similar study in which in vitro inhibition of cyclooxygenase-2 activity was due to active compounds (UA, cis-hydroxycinnamoyl ursolic acid and trans-hydroxycinnamoyl ursolic acid) and cranberry extracts [73]. Forbes and co-workers (2009) synthesized eight derivatives, with 4 showing promising anti-inflammatory and antioxidant activity (50% lipid peroxidation inhibition at 25 µg/mL). The anti-inflammatory activity was tested through the use of enzyme inhibitory assays referred to as in vitro cyclooxegenase-1 (COX-1) and cyclooxegenase-2 (COX-2) [95].

Wei and collaborators (2018) recorded and assessed for anti-inflammatory action the synthesis of 20 UA derivatives comprising oxadiazole, triazolone, and piperazine moieties. Compound (**115**) (Figure 11) displayed the most potent ear inflammation efficacy of all the synthetic compounds (69.76%), which was greater than ibuprofen (25.17%) and indomethacin (26.83%) at 100 mg/kg (i.p.), and was 1- and 2-fold more powerful than conventional medicines. The MTT assay evaluated the cytotoxicity of the compounds and, in contrast to UA, no compounds were reported to show any significant cytotoxic behaviour (IC50 > 100 µmol/L). In addition, the molecular docking findings stated that the UA derivatives showed elevated attraction for effective COX-2 location, potentially exhibiting anti-inflammatory potency through COX-2 inhibition. The findings from this study provide information about UA and its derivatives as anti-inflammatory agents, which could lead to the development of potentially new and safe COX-2 inhibitors [96].

### 4.2. Anticancer Activity

In recent memory, cancer has been ranked as the top deadly diseases and is responsible for many deaths worldwide, being ranked second in economically developed countries. With several forms of cancer being poorly controlled through treatments that have serious side effects themselves and the inevitable limitations of cancer screening programs, chemoprevention and its potential have generated great hope and interest over the past decades [97]. Batra and Sastry (2013) reported isolation of UA (**1**) (Figure 1) from *Ocimum sanctum*, which led to the preparation of three novel derivatives from 3β-acetoxy-urs-12-en-28-oic acid [*N*-(3β-acetoxy-urs-11-oxo-12-en-28-acyl) aniline and [Methyl *N*-(3β-butyryloxyl-urs-12-en-28-oyl)-2-amine acetate] with high in vitro anticancer potency when compared to the ursolic acid parent compound. The modifications of compound **1** occurred at hydroxyl C3, C-11, and carboxylic C-28, as shown in Figure 4. Compounds **2**, **3** and **4** showed potent anti-proliferation activity in cancer cells, with compound **3** displaying higher significance [63].

Meng and colleagues (2017) reported interesting ursolic acid novel derivatives. These authors prepared and synthesized eleven derivatives, which were later assessed for their antitumor inhibition and cytotoxicity against numerous cell lines developed by cancer such as HepG2, cervical carcinoma (HeLa) and BGC-823 using MTT assay. The modifications occurred in positions C-2, C-3 and C-28 to yield 12 compounds (**5**–**17**), as shown in Figure 5. Compound **15** [IC_50_ = 9.25 (HeLa), 21.2 (HepG2) and 8.06 µmol/L (BGC-823)] and compound **16** [IC_50_ = 13.8 (HeLa), 23.7 (HepG2) and 9.15 µmol/L (BGC-823)] were more significant because of their evident high antitumor inhibition against cancer cells. This led to a belief that the antitumor properties of the mentioned compounds could be attributed to the substitution made during the synthesis modification of UA; electron withdrawing, alkanoyloxy imino chain at position C-2, and alkyl side chains at position C-3 [65].

In a different investigation, Shao and co-authors (2011) explored the in vivo and in vitro anticancer activity of a number of synthesized UA derivatives. The synthesized compounds were evaluated against various cancer cells of BGC-823, SH-SY5Y, HELF, HeLa and HepG2 by the assay called MTT. *N*-[3β-acetoxy-urs-12-en-28-oyl]-2-aminodiethanol, *N*-[3b-acetoxy-urs-12-en-28-oyl]-amino-*N*-(2-hydroxyethyl) piperazine, *N*-[3b-acetoxy-urs-12-en-28-oyl]-amino-1-hydroxyethylethoxy piperazine and *N*-[3b-acetoxy-urs-12-en-28-oyl]-amino-4-piperidineethanol exhibited better in vitro antiproliferative activity. The *N*-[3β-acetoxy-urs-12-en-28-oyl]-2-aminodiethanol derivative was reported to have anticancer activity (45 ± 4.3%) on Kunming mice compared to the control group [60]. Rashid and others (2013) produced several derivatives of ursolic acid-triazolyl. The antitumor potency of these compounds was assessed against a number of individual cancer cells, including leukaemia (THP-1), heart (MCF-7), colon (HCT-116), pulmonary (A-549) and ordinary adult epithelial cells (FR-2) using sulforhodamine-B testing. The antitumor potencies of four UA-derivatives against the above-mentioned cells were observed [98]. More authors have reported on ursolic acid and its derivatives as anticancer agents.

In an attempt to develop new anticancer agents, Gu and co-authors (2017) proposed a series of quinoline and oxidiazole derivatives of UA. These researchers evaluated the cytotoxicity of synthesized derivatives against cancer cell lines such as human breast (MDA-MB-231), heptacarcinoma (SMMC-7721), HeLa and normal hepatocyte cell lines (QSG-7701) using MTT assay. The cancer drug atoposide was used as a positive control. The results from this particular study showed interesting significant effects of UA derivatives (**79**–**82**, **91**, **97**) against at least one of the cancer cell lines (MDA-MB-231, SMMC-7721 and HeLa) (IC_50_ < 10µM). Compound **79** showed a higher potency compared to the positive control, with IC_50_ values of 12.49 ± 0.08, 0.36 ± 0.05, and 0.61 ± 0.07 µM against SMMC-7721, HeLa and MDA-MB-231 cell lines, respectively. The bioactive derivatives (**79**–**82**, **91, 97**) did not show any significant cytotoxicity against QSG-7701 (IC_50_ > 40µM) [68].

Ursolic acid caused bax up-regulation and down-regulation of Bcl-2 and discharge from mitochondria of cytochrome C to the cytosol. In addition, UA cleaved caspase-9 and reduced mitochondrial membrane ability. Thus, UA induces apoptosis in MDA-MB-231 cells through both the mitochondrial death pathway and a mechanism dependent on the extrinsic death receptor. Therefore, ursolic acid could be used as a promising anti-cancer drug in treatment of breast cancer [99]. UA (**1**) suppressed the proliferation of androgen-independent DU145 and androgen-dependent human prostate cancer cells (LNCaP) through inhibition of NF-κB and STAT3 activation. Tumour growth was suppressed significantly when four-week-old athymic BALB/c male nude mice inoculated with DU145 cells were treated with (+)-UA (i.p., 200mg/kg, twice a week) for 6 weeks. No significant effects on body weight were observed in mice [100]. Moreover, UA displayed an inhibitor effect through the DNA binding capacity of STAT3, constitutive and inducible STAT3 phosphorylation. In fact, UA therapy also displayed protein expression of cycline D1 and Bcl-2 in a dose-dependent way. In the G1 phase, UA quickly collected in the cell population at 50 mM, sucked by flow cytometry, confirming that ursolic acid caused G1 in the regulation of the cell cycle [101].

Many clinical trials using chemical, subcutaneous, orthotopic human xenograft and randomly transgenic tumour development designs have provided extensive proof that naturally existing and synthetic UA products have chemopreventive and therapeutic effects. Liposomal ursolic acid (LUA) was used as a fresh drug in normal young volunteers and in individuals with developed solid tumours in order to determine the highest permitted level (MTD), dose-limiting toxicities (DLTs), and UA pharmacokinetics. A total sample of LUA (11, 22, 37, 56, 74, 98 and 130 mg/m2) was given to 63 subjects (4 patients, 35 healthy participants and 24 adults). For the first season, clinical information revealed that LUA had manageable 98 mg/m^2^ MTD toxicity. The DLTs were predominantly diarrhoea and hepatotoxicity. Furthermore, a linear pharmacokinetic profile was found in the ursolic acid liposomal model [52]. UA is a promising antitumor agent. It can induce apoptosis in tumour cells, on one hand, and stop ordinary cell transformation, on the other. It also interferes with countless proteins, including those that directly serve the structure of DNA. It can prevent the development of many tumour cell types and cause apoptosis. It has been shown that this compound acts at different phases of tumour growth. It efficiently prevents angiogenesis, tumour cell intrusion, and metastasis. It is comparatively non-toxic and can be used in clinical practice as a chemopreventive/chemoprotective product [102]. Therefore, it is evident that UA and its analogues are promising therapeutic agents against several types of cancers.

### 4.3. Antibacterial

Public health is faced with many challenges, including the resistance of many already-available antibiotics. There is hope in the use of naturally occurring products as a substitute that can produce a better and more promising therapeutic effect through the discovery of new compounds and derivatives against bacterial pathogens. Nascimento and colleagues (2014) semi-synthesized two analogues of UA isolated from the *Sambucus australis* plant. Modifications of the UA structure were made at C-3 (hydroxyl group) to yield 3β-formyloxy-urs-12-en-28-oic acid (**19**) and 3β-acetoxy-urs-12-en-28-oic acid (**20**), as shown in Figure 5. Ursolic acid and its analogues showed significant antibacterial and antioxidant properties. These authors tested the minimal inhibitory concentration (MIC) of both ursolic acid and its derivatives against several microbial pathogen strains (*Staphylococcus aureus, Bacillus cereus, Shigella flexneri, two strains of Escherichia coli, Aeromonas caveae, Pseudomonas aeruginosa, Klebsiella pneumonia, and Vibrio colareae*) using the microdilution method. Compound **19** (64 μg/mL) when combined with kanamycin showed significant activity towards Escherichia coli compared to the multidrug resistance observed from sputum, which reduced MIC from 128 μg/mL to 8 μg/mL [64]. This shows that the antibacterial efficiency of UA improves with modification.

Wang and co-authors (2016), in their investigation, explored the biological antibacterial properties of UA against *Staphylococcus aureus* (MRSA), which is resistant to methicillin. Researchers in this research noted that with no haemolytic consequences, UA produced a reduction in staphylococcal membrane integrity. UA-treated protein type relative to ordinary MRSA cells showed that UA impacted translation efficiency, chaperon subunits, ribonuclease, oxidative reactions, and glycolysis of a variety of proteins engaged in the transformation method [8]. Zhao and others (2018) isolated ursolic acid from the leaves of *Ilex hainanensis* Merr. These authors assessed seven UA derivatives against *Fusobacterium nucleatum* (Gram-ve) and *Streptococcus mutans* (Gram + ve) for their antibacterial activity. Only three derivatives displayed important activity against Gram positive at distinct levels and displayed substantial activity against Gram negative at a minimum inhibitor level (MIC) of 625 μg/mL [103]. Park and other researchers (2015) recorded the action of ursolic acid (UA) against bacterial (*Streptococcus mutans*) development in their quest for powerful antimicrobial drugs [104].

### 4.4. Anti-Diabetes

Diabetes mellitus is a chronic disease that is triggered by insulin resistance or deficiency. This disease is one of the major risks to human well-being worldwide [105]. Long-term use of oral hypoglycaemic agents may reduce their pharmacological activity. Due to an utter glucose insufficiency, type 1 diabetes mellitus (DM1) results in hyperglycaemia. This results in numerous problems, including microvascular and macrovascular modifications in pathology, retinopathy, neuropathy, diabetic nephropathy, diabetic osteopenia, and osteoporosis. Recent clinical studies have shown that DM1 increases knee, vertebral, proximal humerus, tibia, shoulder and knee fracture hazards independent of bone mineral thickness (BMD) [71]. Consequently, there is a growing interest in scientific knowledge regarding, and clinical approval for the search for and use of, antidiabetic drugs from natural sources. Ursolic acid is known to have a beneficial impact on lowering blood glucose concentrations and on the healing of diabetic problems in diabetic mice. On the other hand, derivatives of UA reduce protein tyrosine phosphatase 1B, improve phosphorylation of insulin receptors, and stimulate glucose absorption [69].

Wu and colleagues (2014) produced a series of UA derivatives and evaluated their potency as an antidiabetic agent against α-glucosidase. Most of the derivatives showed activity, except for one compound with four derivatives showing significant inhibition at IC_50_ of 2.66 ± 0.84, 1.01 ± 0.44, 3.26 ± 0.21 and 3.24 ± 0.21 µM [106]. Wu and co-authors (2015) explored the antidiabetic activity of several UA analogues against the α-glucosidase. These authors discovered that most of these analogues exhibited significant inhibition activity, with the two highest potencies being IC_50_ = 1.27 ± 0.27 and IC_50_ = 1.28 ± 0.27 µM, compared to the other derivatives and the positive control [67]. Khusnutdinova and others (2015) explored UA derivatives with respect to their medicinal effects (in vitro inhibition) against α-glucosidase. 2,3-Indole UA derivative (**107**) showed a greater efficiency against α-glucosidase with an IC_50_ value of 115.1 μM; 3,5 times more potent than the standard drug (acarbose) [69].

Yu and others (2015) reported the antidiabetic effects of UA derivatives in bone deteriorations (BMD) of diabetic mice (6 weeks old) induced by streptozotocin (STZ). In this research study, UA derivative (**108**) was used as hypoglycaemic agents to evaluate their therapeutic effects against non-obese type 2 diabetic mice for two weeks. In this study, biomarkers in serum and urine were measured. In addition, protein expression, gene, and histomorphology analysis were measured from mice tibias. Moreover, femurs were taken for bone Ca measurement and trabecular bone three-dimensional architecture. These authors revealed a reduced testosterone level in the STZ serum of mice. UA analogues showed increased bone Ca, BMD, significantly increased Fibroblast growth factor 23 (FGF-23) and osteocalcin (OCN), and reduced diabetic mice parathyroid hormone (PTH) levels and crosslaps (CTX). UAD reversed the trabecular deleterious effects caused by STZ and stimulated remodelling of the bone. Treatment with UAD for the STZ group significantly increased the osteoprotegerin (OPG)/nuclear factor (NF)-κB ligand (RANKL) ratio. It has been shown that UA derivatives can improve STZ-induced bone deterioration by improving the dysfunction of mesenchymal stem cells [71].

Chinese hamster ovary (CHO-K1) neurons with the TGR5 gene were transfected by Lo and others (2017) to test the antidiabetic effects of UA. Using a fluorescent marker, the features of the transfected cells were verified through glucose uptake. In addition, NCI-H716 cells that secreted incretin were also explored, and ELISA sets were used to quantify the glucagon-like protein (GLP-1) concentrations. In fact, Type 1-like diabetic rats induced by STZ were used to define the impact of in vitro ursolic acid. The level of UA dependently enhanced glucose uptake in TGR5 producing CHO-K1 cells. UA caused a concentration-dependent increase in GLP-1 secretion of NCI-H716 cells, which was inhibited by triamterene at efficient levels to inhibit TGR5. Ursolic acid also improved the amount of GLP-1 plasma by activating TGR5, which was further described in vitro with diabetic rats of type 1 [107].

### 4.5. Neuroprotective Activity

Neurological disorders include anxiety, depression, stroke and Alzheimer’s disease, among others [64]. Long-term neurological outcomes cannot be improved by subarachnoid haemorrhage (SAH) therapies that reduce the incidence of cerebral vasospasm, indicating that their importance in patient outcome has been misinterpreted. More recently, early brain injury (EBI) was seen as the primary cause of negative results in patients with SAH, rather than cerebral vasospasm. UA remains an important and popular antioxidant reagent, with many reports suggesting that it has brain protective effects against ischemic stroke [108]. Oxidative stress was identified as being among the complicated causes associated with neuronal death after SAH. Reactive oxygen species (ROS) or reactive nitrogen species (RNS), such as peroxynitrite (ONOO^−^), nitric oxide (NOO), superoxide anion (•O_2_), hydroxyl radical (•OH), and hydrogen peroxide (H_2_O_2_), play a crucial role after SAH [109]. Several researchers have shown that redox-sensitive Nrf2 activation has a central position in improving the endogenous defence mechanism through which the brain protects itself against ischemic harm and recovers from the stroke [110].

Sahni and co-workers (2016) reported that UA derivatives of C-6 (C-17 propyl amide) and C-2 (C-3 methyl ester) displayed important neuroprotective activity in an in vivo model of D-galactose-induced neurotoxicity in rats. Thus, C-2 and C-6 could be advantageous in cognitive disorder therapy, for instance, in the treatment of Alzheimer’s disease and dementia [66]. Li and colleagues (2012) reported an interesting piece of research on the in vivo neuroprotective effect of UA on cerebral dischemia. These researchers showed important and necessary anti-inflammatory and antioxidative effects of UA in the brains of the mice after middle cerebral occlusion (MCAO) at 24 h [110]. Moreover, Zhang and colleagues (2014) proposed in vivo effects of UA against EBI (neurological deficiency, blood–brain barrier disruption, neural cell apoptosis, and brain edema) after SAH on Sprague Dawley rat models. The effects of UA in reducing EBI could be explained by alleviation of oxidative stress on SD rats. Therefore, UA might be regarded as a potential therapeutic agent for neurological disorders [108].

Another disease, known as Parkinson’s disease (PD), is a chronic progressive neurodegenerative disorder that is defined both by motor and nonmotor characteristics. Through its progressive degenerative impacts on mobility and muscle control, it has a major clinical effect on patients, families, and caregivers [111]. PD is the second most common age-related progressive neurodegenerative disease, following Alzheimer’s, and characterized by dopaminergic (DA) neuron reduction, α-synuclein protein accumulation, and neuroinflammation [112,113,114]. Rai and others (2015) explored the neuroprotective efficiency of UA in 1-methyl-4-phenyl-1,2,3,6-tetrahydropyridine (MPTP)-induced PD mouse models (5 mg/kg, 25 mg/kg, and 50 mg/kg body weight). Immunostaining of substantia nigra dopaminergic cells was also conducted, as well as HPLC quantification of dopamine and its 3,4-dihydroxyphenylacetic acid (DOPAC) and homovanilic acid (HVA) metabolites. In addition, these authors discovered that UA increases cognitive deficits, restores modified dopamine levels (*p* < 0.001), and protects the MPTP-intoxicated mouse’s dopaminergic neurons (*p* < 0.01). Among three distinct doses, the most efficient dose for PD was 25 mg/kg body wt. [113]. The literature suggests that UA and its derivatives can be considered to be potential therapeutic agents for neurological disorders.

### 4.6. Herbicidal Activity

Compounds with allelophatic potency are denoted as allelochemicals. Allelochemicals can cause inhibition of photosynthesis, decreases in chlorophyll content, inhibition of enzymatic activity, and disruption of the membrane and structure of cells. These effects can also be beneficial for host plants, inhibiting the development of pathogenic fungus organisms as a result of their antibacterial, antifungal and growth-inhibiting activities [115]. Phenolic compounds and their derivatives, particularly simple phenols such as phenolic acids, affect the permeability of the membrane, inhibiting the growth of the aerial part and the elongation of the roots [89]. Specifically, the IC_50_ values of UA range between 75 and 700 μM for drastic growth of *Lactuca sativa* (lettuce) [89,116]. Tuyen et al. [74] explored an interesting isolation and elucidation of a naturally occurring UA derivative called 2_,3_,7_,23-tetrahydroxyurs-12-ene-28-oic acid (**118**) from *Castanea crenata* (Japanese chestnut). These authors reported the herbicidal effects of compound **118** in root and shoot growth of *Echinochloa crus-galli* (barnyard grass), *L. sativa*, and *Raphanus sativus* (radish) p-hydroxybenzoic acid. Compound **118** inhibition (IC_50_ = 2.62 and 0.41 mM) was >5 times higher than p-hydroxybenzoic acid (IC_50_ = 15.33 and 2.11 mM) in the *Echinochloa crus-galli* shoot and root growth, respectively. These results indicate that the isolated compound **118** has the ability to create natural herbicides for *Echinochloa crus-galli* management [74].

The elder *Sambucus nigra* L. is a major berry crop plant globally that has been commonly converted into jams, jellies and drinks commonly used in gastronomy, and which has previously been used for its prospective health advantages [117,118]. Basas-Jaumandreu et al. [118] detected ursolic acid in the leaves and inflorescences of *S. nigra* by means of Gas Chromatography with Electron Impact Mass Spectrometry (GC-EIMS). UA (allelochemical) was most abundant in the leaves and flowers of *S. nigra* (57 and 67%, respectively) [118]. In addition, Saidi et al. [115] studied the herbicidal activity of triterpenoids isolated and elucidated UA from ethyl acetate flower extract of *Citharexylum spinosum* L. including UA in germination and seedlings growth of plants such as *R. sativus, L. sativa* and *Phalaris canariensis* (canary grass). The roots were the most adversely affected; UA (**1**) was the most phytotoxic and caused 91.74 ± 0.24% and 89.55 ± 0.31% inhibition of roots and shoots, respectively [115].

UA was isolated and characterized from the *Salvia syriaca* L. methanol extract. Significant growth inhibition (GI) of wheat seedlings was shown by UA, decreasing shoot lengths and *Triticum aestivum* cv (wheat) seedling roots when evaluated at 0.2 mg/L (200 ppm) as reported by Abu-Irmaileh and Abu-Zarga (2015). These authors reported UA GI (*p* = 0.05) on seed germination, root length and shoot height of *T. aestivum* at 70%, 9.1 and 6 cm, respectively [119].

## 5. Conclusions

It is evident that the therapeutic activity of ursolic acid improves with modifications when compared with various current used standard drugs. A majority of reports described increases in the solubility, bioavailability, and potency of UA through modifications at different positions. Many authors report modifications at the hydroxyl position C-3, hydrogen C-2, and carboxylic acid position C-28. Although several synthetic ursolic acid inhibitors have already been submitted, fresh derivatives still need to be developed and synthesized to further enhance their medicinal impacts on non-communicable diseases. Therefore, UA and its derivatives provide hope as potential therapeutic agents for non-communicable diseases through in vivo, in vitro, preclinical and clinical trials. Moreover, UA and its derivatives possess herbicidal effects in the growth of various plants.

## Figures and Tables

**Figure 1 molecules-24-02751-f001:**
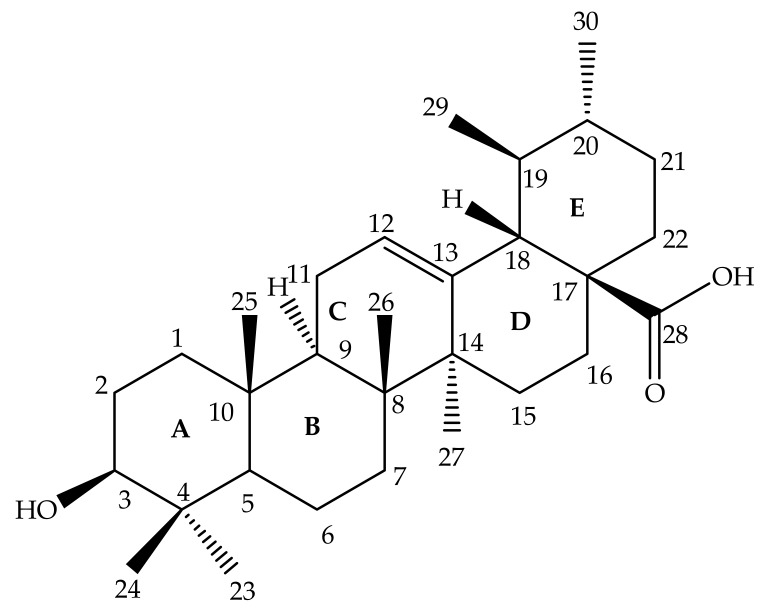
Ursolic acid chemical structure.

**Figure 2 molecules-24-02751-f002:**
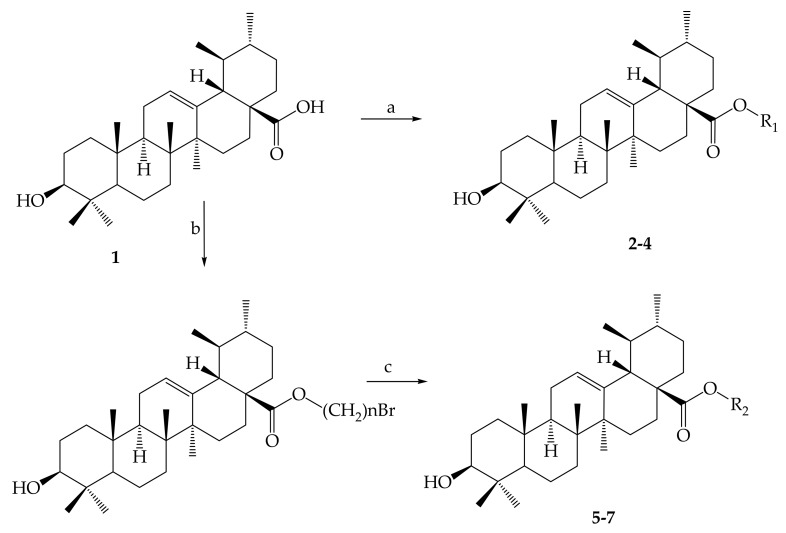
Preparation of UA ester derivatives. Reagents and conditions: (a) CH_3_(CH_2_)_n_Br, potassium carbonate (K_2_CO_3_), dimethylformamide ((CH₃)₂NCH); (b) Br(CH_2_)_n_Br, K_2_CO_3_, (CH₃)₂NCH; (c) Acetonitrile CH_3_CN/silver nitrate (AgNO_3_), COO(CH_2_)_n_Br. 2: R_1_ = CH_2_CH_3_; 3: R_1_ = (CH_2_)_2_CH_3_; 4: R_1_ = (CH_2_)_3_CH_3_; 5: R_2_ = (CH_2_)_2_ONO_2_; 6: R_2_ = (CH_2_)_3_ONO_2_; 7: R_2_ = (CH_2_)_4_ONO_2_.

**Figure 3 molecules-24-02751-f003:**
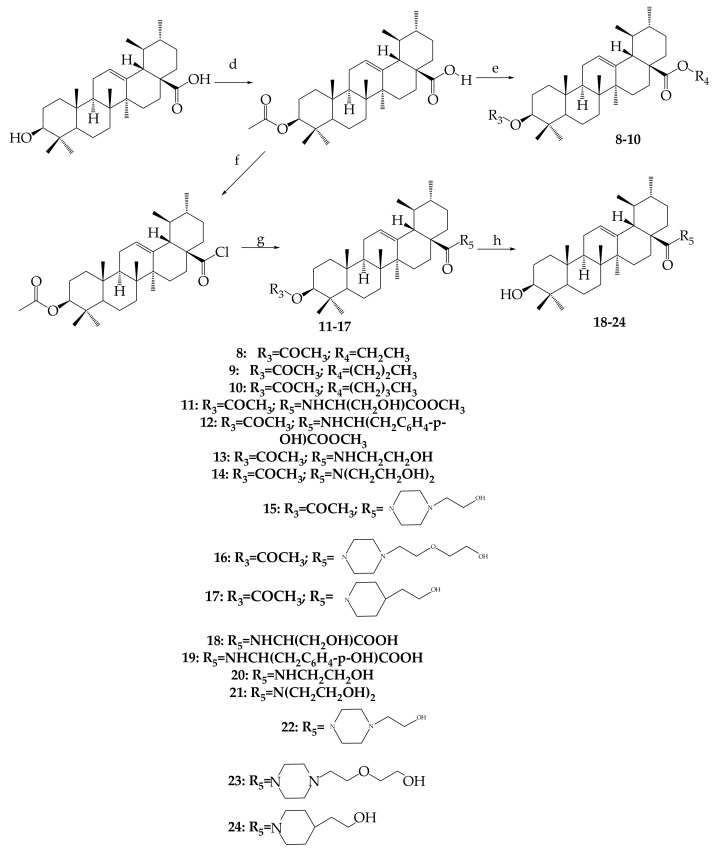
Modification of UA yielded amides and ester derivatives. Reagents and conditions: (d) anhydride/Pyridine/*N*, *N*-dimethyl-4-aminopyridine (DMAP); (e) CH_3_(CH_2_)_n_Br, K_2_CO_3_, (CH₃)₂NCH; (f) rhodium carbonyl chloride ((CO)_2_Cl), dichloromethane (CH_2_Cl_2_); (g) CH_2_Cl_2_, triethylamine (Et_3_N), HR; (h) NaOH, methanol (CH_3_OH)/tetrahydrofuran (THF). All the reactions were carried out at room temperature.

**Figure 4 molecules-24-02751-f004:**
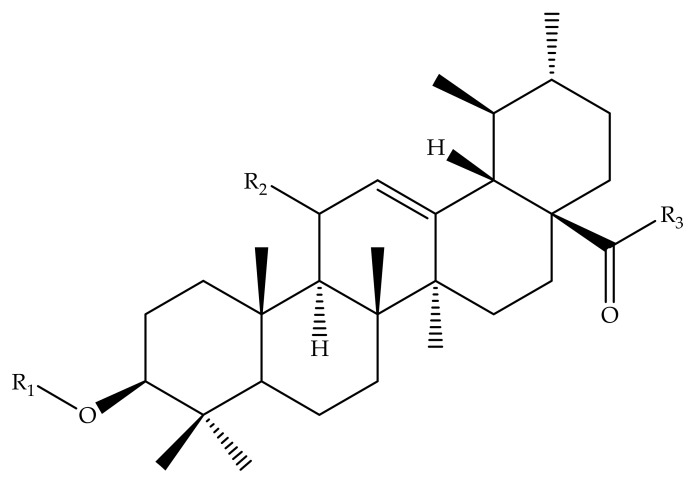
Some of UA derivatives. 1, 25–27; 1: R_1_ = H; R_2_ = H; R_3_ = OH; 25: R_1_ = OCCH_3_; R_2_ = H; R_3_=OH; 26: R_1_ = OCCH_3_; R_2_ = O; R_3_ = NHC_6_H_5_; 27: R_1_ = OCH_2_CH_2_CH_3_; R_2_ = H; R_3_ = NHCH_2_CO_2_CH_3_.

**Figure 5 molecules-24-02751-f005:**
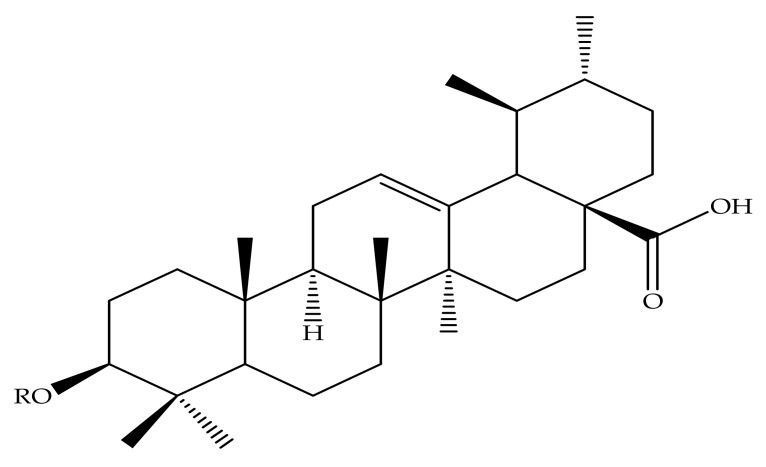
UA analogues. 28–29; 28: R = COH; 29: R = COCH_3_.

**Figure 6 molecules-24-02751-f006:**
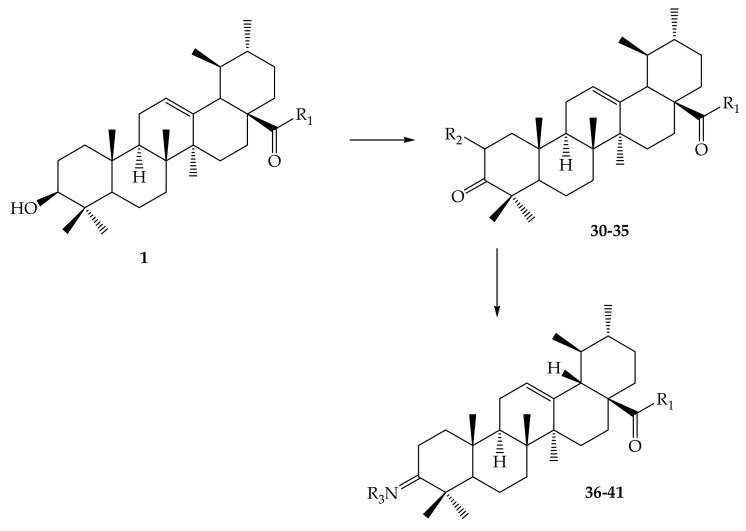
Derivatives synthesized from UA.

**Figure 7 molecules-24-02751-f007:**
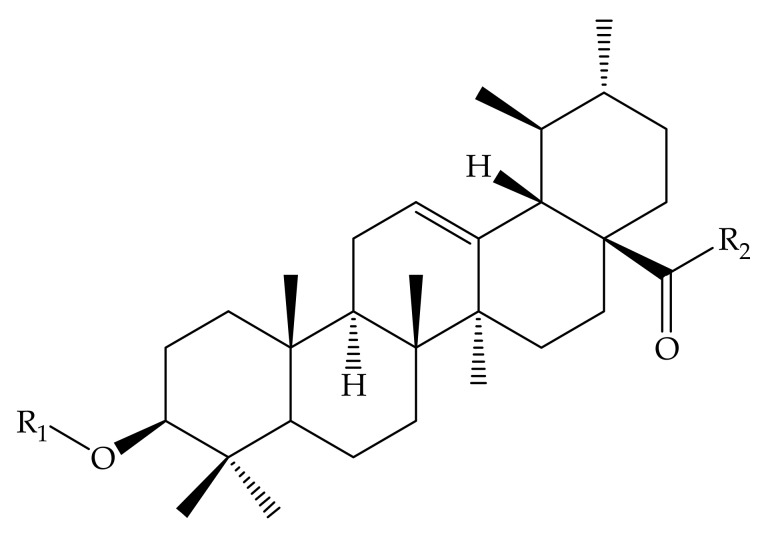
Synthesized UA derivatives. 42–46; 42: R_1_ = COCH_3_; R_2_=OH; 43: R_1_ = COCH_2_CH_3_; R_2_ = OH; 44: R_1_ = COCH_2_CH_2_CH_3_; R_2_ = OH; 5: R_1_ = H; R_2_ = OCH_3_; 46: R_1_ = H; R_2_ = NHCH_2_CH_2_CH_3_.

**Figure 8 molecules-24-02751-f008:**
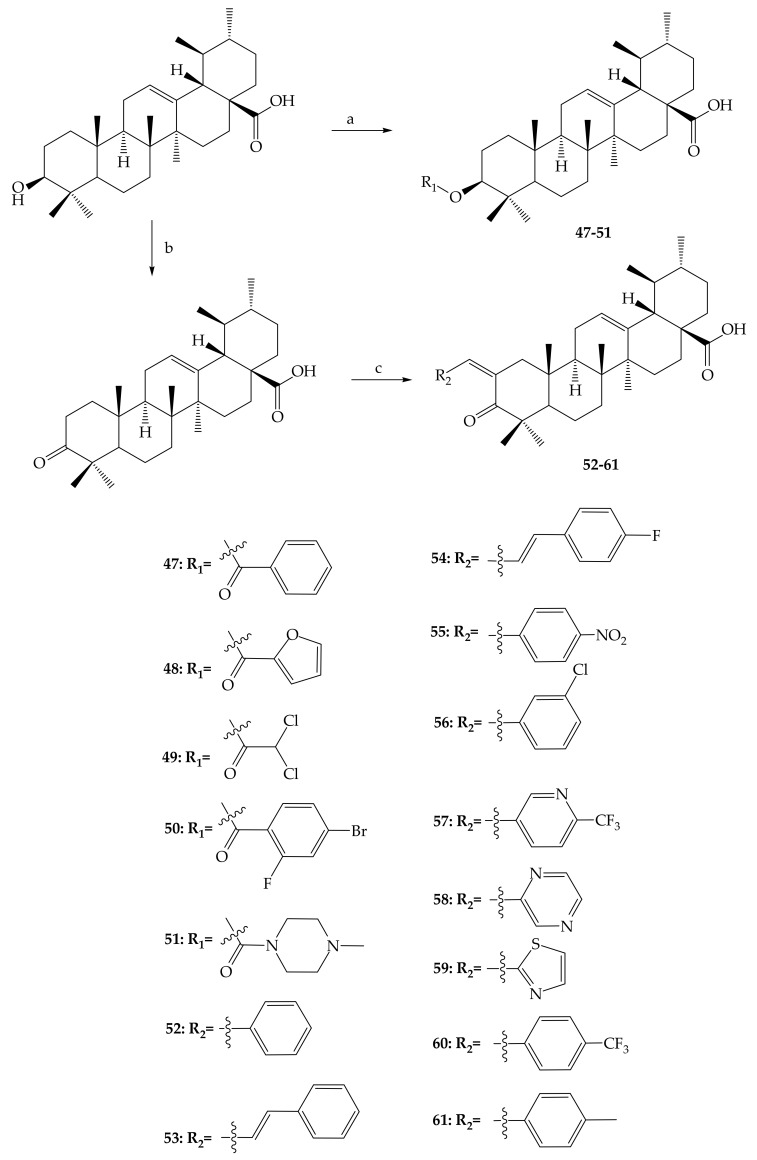
Synthesis of UA derivatives; reagents and conditions: (a) acid chloride or anhydride, DMAP, pyridine, reflux; (b) chromium (III) oxide (CrO_3_), sulphuric acid (H_2_SO_4_), acetone, 0 °C, 1 h; (c) R1-CHO, potassium hydroxide (KOH), ethanol, room temperature.

**Figure 9 molecules-24-02751-f009:**
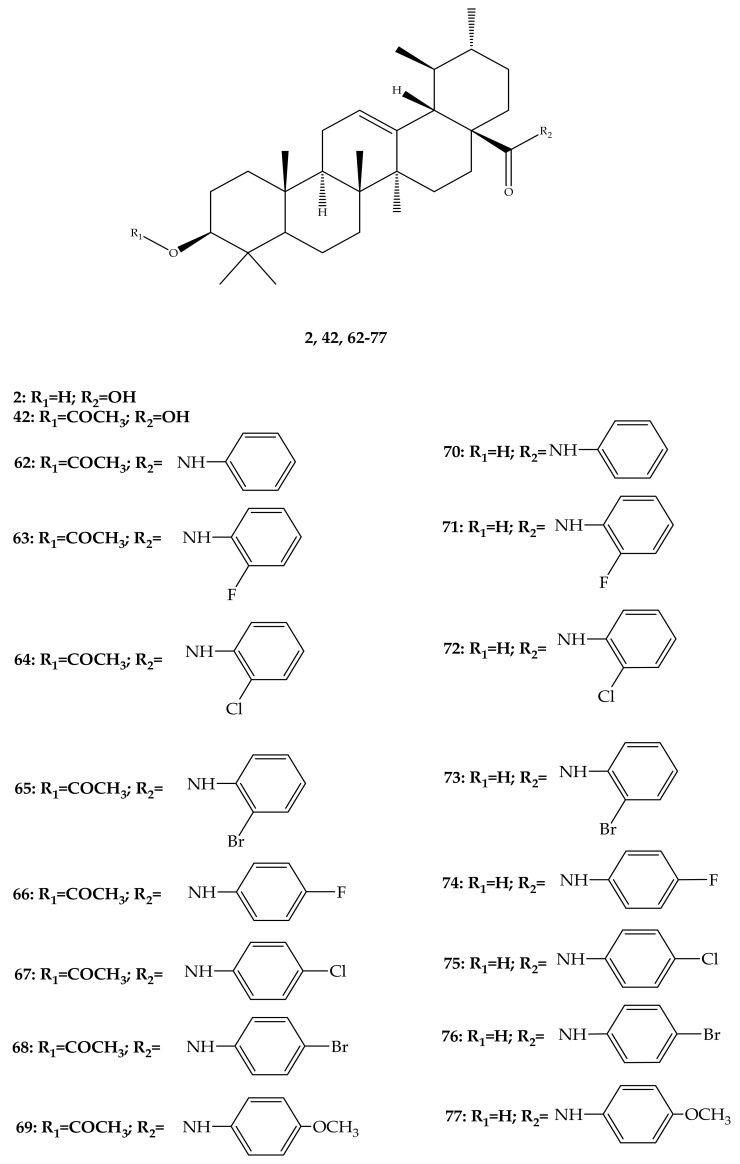
Some reported UA analogues.

**Figure 10 molecules-24-02751-f010:**
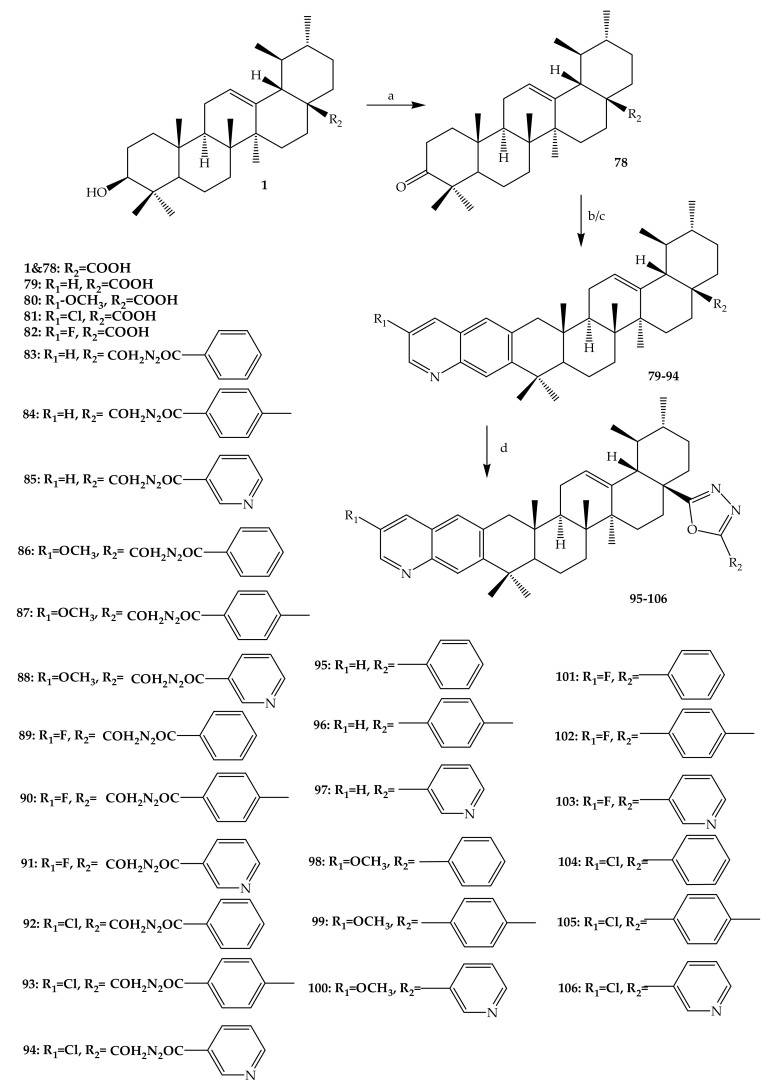
Summary of reaction scheme to yield UA analogues (**78**–**106**). Reagents and conditions: (a) Jones reagent, acetone at 0 °C for 5 h; (b) ethanol, substituted oaminobenzaldehyde, potassium hydroxide (KOH), reflux under N_2_ atmospheric conditions for 24 h; (c) Thionyl chloride (SOCl_2_), benzene, reflux for 3 h; RCONHNH_2_, trimethylamine (Et_3_N), dichloromethane (CH_2_Cl_2_)/ether at room temperature for 8–12 h; (d) p-Toluenesulfonic acid (TsOH), toluene, reflux for 6 h.

**Figure 11 molecules-24-02751-f011:**
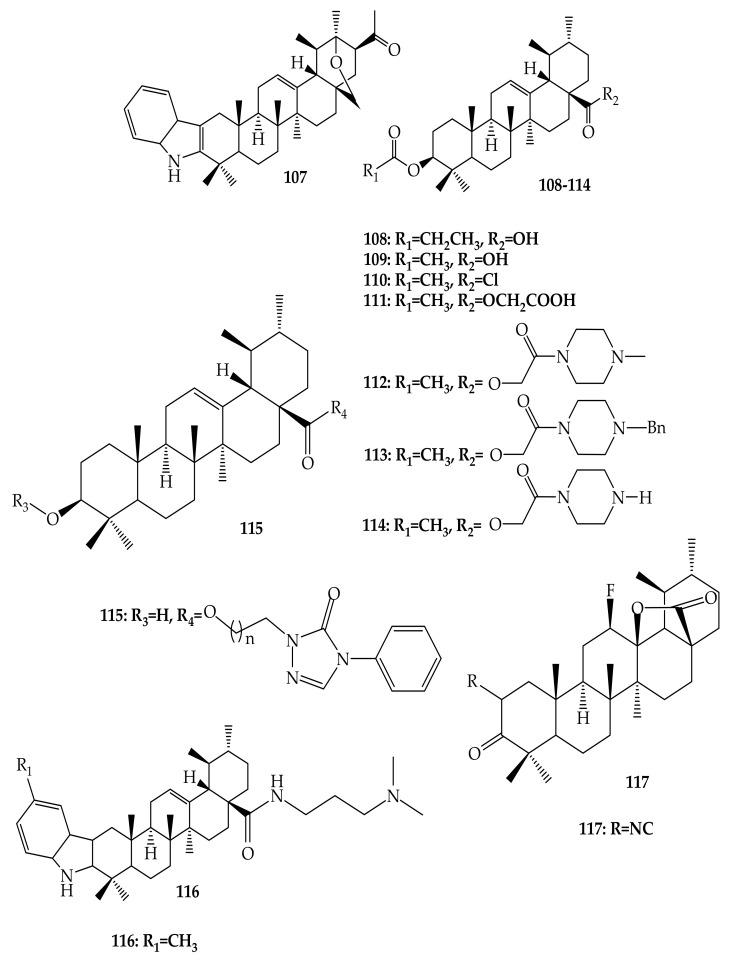
Some derivatives prepared by modification of UA with interesting potential as therapeutic agents.

**Table 1 molecules-24-02751-t001:** Pentacyclic triterpene compounds from plant sources with their chemical characteristics.

Pentacyclic Triterpenes	Compound	Chemical Formula	Molecular Mass (g/mol)	Reference(s)
ursane
	ursolic acid	C_30_H_48_O_3_	456.71	[7,10,11,12,13,14,15]
	uvaol	C_30_H_50_O_2_	442.72	[16,17,18]
	α-amyrin	C_30_H_50_O	426.70	[17,18,19,20]
oleanane	oleanolic acid	C_30_H_48_O_3_	456.71	[13,14,15,20,21,22,23]
	maslinic acid	C_30_H_48_O_4_	472.70	[20,22,24]
	β-amyrin	C_30_H_50_O	442.70	[18,20,25]
	erythrodiol	C_30_H_50_O_2_	442.72	[17,18,20]
lupane	lupeol	C_30_H_50_O	426.70	[15,20,26,27,28,29,30,31,32,33,34]
	betulin	C_30_H_50_O_2_	442.72	[15,17]
	betulinic acid	C_30_H_48_O_3_	456.71	[15,17,26,35,36,37]
dammarane	dammarane	C_30_H_54_	414.75	[38,39,40]
	pseudojujubogenin-3-o-β-d-glucopyranoside	C_36_H_58_O_10_	650.85	[41]
	hopane	C_30_H_52_	412.75	[38,42]
	diploptene	C_30_H_50_	410.73	[43,44]
	bacteriohopanetetrol	C_35_H_62_O_4_	546.89	[43,45]
sterols	cholesterol	C_27_H_46_O	386.65	[46]
	ergosterol	C_28_H_44_O	396.65	[43]
	β-sitosterol	C_29_H_50_O	414.71	[47,48,49,50]

**Table 2 molecules-24-02751-t002:** Some novel analogues derived, corresponding with Figure 6, above.

Compound Number	R_1_	R_2_	R_3_
**1**	OH	-	-
**30**	OCH_3_	CHO	-
**31**	OC_2_H_5_	CHO	-
**32**	OCH(CH_3_)_2_	CHO	-
**33**	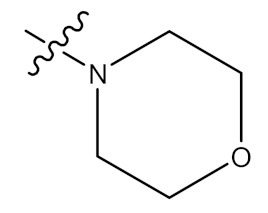	CHO	-
**34**	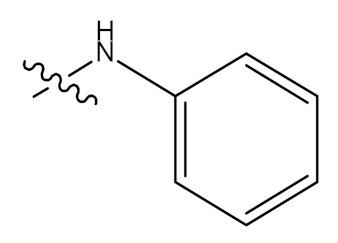	CHO	-
**35**	OCH_2_CH_2_CH_3_	CHO	-
**36**	OH	-	OCOCH_3_
**37**	Cl	-	OCOCH_3_
**38**	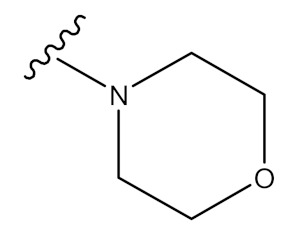	-	OCOCH_3_
**39**	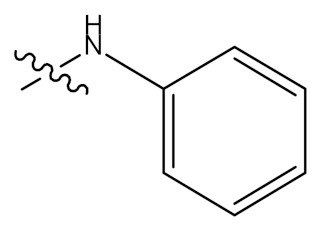	-	OCOCH_3_
**40**	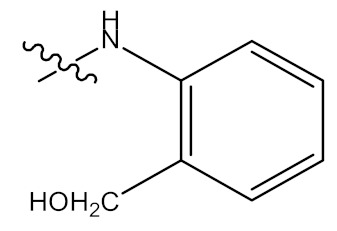	-	OCOCH_3_
**41**	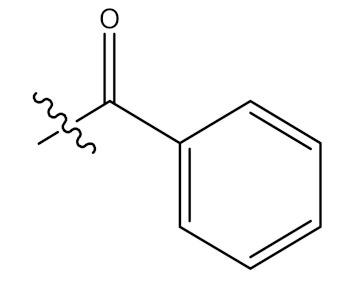	-	OCOCH_3_

**Table 3 molecules-24-02751-t003:** Sources of UA with their biological properties.

Plants Species (Family)	Plant Part (Solvent Crude Fraction)	UA Content (mg or g)	Type of Study	Biological Effects	Reference(s)
*Fragrae fragrans* (Gentianaceae)	fruits (methanol)	91 g	in vitro	antiproliferation	[14]
*Saurauja roxburghii* (Actinidiaceae)	leaves (methanol)	nr	in vitro	cytotoxicityagainst glioma cells	[10]
*Ocimum sanctum* (Lamiaceae)	whole plant (methanol, acetone, acetonitrile and ethyl acetate)	11.21 mg	in vitro	anticancer and antiproliferation	[63,87,88]
*Eucalyptus* (Myrtaceae)	leaves (acetone)	nr	in vivo	neuro-protective agent	[66,75,89]
*Malus pumila* (Rosaceae)	fruits	nr	in vitro	antitumor	[79]
*Tribulus arabicus* (Zygophyllaceae)	aerial parts (ethanol)	1 g	in vitro and in vivo	antihyperuricemic activity	[90]
*Panax ginseng* (Araliaceae)	roots	nr	in vivo	antihypertensive, antihyperlipidemic and antioxidant effects	[91]
*Bursera cuneata* (Burseraceae)	aerial parts (dichloromethane)	33.3 mg	in vitro and in vivo	anti-inflammatory and antihistaminic activity	[92]
*Sambucus australis* (Adoxaceae)	aerial parts (ethanol)	180 mg	in vitro	antibacterial and antioxidant	[64]


nr = not reported.

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
