# Peer review of "Ursolic Acid and Its Derivatives as Bioactive Agents"

_molecules, 2019, doi:10.3390/molecules24152751_

Round 1
Reviewer 1 Report
The manuscript in reference compiles the available information regarding the chemistry, sources and biological activity of ursolic acid (UA) and some derivatives. The manuscript is interesting, well-written and has important information about this natural compound, deserving publication in molecules. However, some minor points must be addressed prior acceptance.
Some mistakes in grammar and style are found within manuscript. A detailed scrutiny can improve this fact.
The abstract seems to be an introduction rather than an abstract. I recommend refocusing the abstract towards the aim and scope of the review regarding the UA features.
The first part of introduction is vague, because it is described general information but this does not contribute to the information regarding UA. I suggest a refocusing only to the triterpenoids and the precise biological effects. Why is not mentioned something about on-communicable diseases in introduction?
Regarding chemistry of UA, why authors selected those transformations and derivatives? In literature, there are more examples of UA derivatives and there is no reason to include only these in the manuscript. In this regard, an explanation can thus be included to support such inclusions.
I consider the biological effects are poorly compiled and explored, since there are more examples in literature of biological effects of UA and derivatives, even some clinical trials. I think it is important to include more information and describe it in detail. For instance, neuroprotection of UA had been related to its antioxidant and anti-inflammatory effects (e.g., 10.1155/2019/8512048) and this biological effect is described in four lines.
The last sentence of the conclusion section is very general and can be oriented to a more specific conclusion regarding UA and derivatives.
Author Response
Response 1: The minor points about the review entitled “Ursolic acid and its derivatives as bioactive agents” were addressed as suggested by reviewer 1.
Response 2: The grammar mistakes within manuscript were corrected as indicated.
Response 3: The abstract was re-written in order to fulfill the aim and the scope of the manuscript as suggested.
Response 4: The first part which was not related to UA was completely removed from the review and the information about non-communicable diseases was added with precise biological potency of pentacyclic triterpenoids.
Response 5: More examples of UA derivatives were added to increase the quality of the review. The authors specifically chose some of the bioactive derivatives with therapeutic effects against non-communicable diseases reported in the last decade.
Response 6: More examples of biological potency and some clinical trials of UA its derivatives were added especially neuroprotection and antidiabetic studies to enhance the quality of the manuscript. Most importantly the information about UA neuroprotective, antioxidant and anti-inflammatory effects was linked in the review manuscript. The last line was added to highlight the overall therapeutic importance of UA and derivatives.
Reviewer 2 Report
The authors of the manuscript (Mole.-542390) reviewed the recent advances in ursolic acid and its derivatives, including their isolation/preparation and bioactivities. Actually, studies on triterpenoids are a quite hot area because more than 50 reviews on the related topics have been published since 2000 (Scifinder, key words: Ursolic Acid; Derivatives; review) and the latest ten papers have been listed below. But it seems that no one of them has been cited in the manuscript.
1. Zou, Junjie; Lin, Juanfang; Li, Chao; Zhao, Ruirui; Fan, Lulu; Yu, Jesse; Shao, Jingwei. Ursolic Acid in Cancer Treatment and Metastatic Chemoprevention: From Synthesized Derivatives to Nanoformulations in Preclinical Studies. Current Cancer Drug Targets (2019), 19(4), 245-256.
2. Ren, Yulin; Kinghorn, A. Douglas. Natural Product Triterpenoids and Their Semi-Synthetic Derivatives with Potential Anticancer Activity. Planta Medica (2019), Ahead of Print.
3. Sultana, Nighat. Triterpenes and triterpenoids clinically useful with multiple targets in cancer, malaria and more treatment: focus on potential therapeutic value. International Journal of Biochemistry Research & Review (2017), 16(2), 1-35.
4. Kataev, V. E.; Khaybullin, R. N.; Garifullin, B. F.; Sharipova, R. R. New Targets for Growth Inhibition of Mycobacterium tuberculosis: Why Do Natural Terpenoids Exhibit Anti- tubercular Activity? Russian Journal of Bioorganic Chemistry (2018), 44(4), 438-452.
5. Peron, Gregorio; Marzaro, Giovanni; Dall'Acqua, Stefano. Known Triterpenes and their Derivatives as Scaffolds for the Development of New Therapeutic Agents for Cancer. Current Medicinal Chemistry (2018), 25(10), 1259-1269.
6. Lopez-Hortas, Lucia; Perez-Larran, Patricia; Gonzalez-Munoz, Maria Jesus; Falque, Elena; Dominguez, Herminia. Recent developments on the extraction and application of ursolic acid. A review. Food Research International (2018), 103, 130-149.
7. Yin, Ran; Li, Tong; Tian, Jing Xin; Xi, Pan; Liu, Rui Hai. Ursolic acid, a potential anticancer compound for breast cancer therapy. Critical Reviews in Food Science and Nutrition (2018), 58(4), 568-574.
8. Hussain, Hidayat; Green, Ivan R.; Ali, Iftikhar; Khan, Ikhlas A.; Ali, Zulfiqar; Al-Sadi, Abdullah M.; Ahmed, Ishtiaq. Ursolic acid derivatives for pharmaceutical use: a patent review (2012-2016). Expert Opinion on Therapeutic Patents (2017), 27(9), 1061-1072.
9. Huang, Qiuxia; Chen, Hongfei; Ren, Yuyan; Wang, Zhe; Zeng, Peiyu; Li, Xuan; Wang, Juan; Zheng, Xing. Anti-hepatocellular carcinoma activity and mechanism of chemopreventive compounds: ursolic acid derivatives. Pharmaceutical Biology (Abingdon, United Kingdom) (2016), 54(12), 3189-3196.
10. Kashyap, Dharambir; Sharma, Ajay; Tuli, Hardeep S.; Punia, Sandeep; Sharma, Anil K. Ursolic Acid and Oleanolic Acid: Pentacyclic Terpenoids with Promising Anti-Inflammatory Activities. Recent Patents on Inflammation & Allergy Drug Discovery (2016), 10(1), 21-33.
Furthermore, the manuscript has not been well written. Too many grammatical mistakes in the main body of it (too many to be pointed out one by one!), especially in the first part make it more like a draft and tough to be followed. Some key info of a number of cited references has not been included as well. Based on above, I recommender rejection of the manuscript.
Author Response
Response 1: The recent information about UA and its derivatives from the suggested articles were added and cited to improve the quality and relevance of this review. The UA and derivatives are mostly talked about these days; however, it is necessary to give an update with recent information about current modifications and clinical trials.
Response 2: The grammar errors were comprehensively scrutinized and addressed to improve the quality and relevance of the review. The key information about UA and its derivatives have been included as suggested.
Reviewer 3 Report
The manuscript entitled “Ursolic Acid and its derivatives as bioactive agents”, presents a good scientific and even technical contribution to the state of the art of a bioactive compounds with high demands in pharmaceutical and food industries actually.
Moreover, this review manuscript is well written and updated and offers a good contribution on the pentacyclic terpenoids and its applications on non-communicable deseases. This manuscript deserves to be published.
Author Response
Response 1: Thank you and new information was added to make the review more readable.